# WHEN IS MULTILINGUALITY A CURSE?
# LANGUAGE MODELING FOR 250 HIGH- AND LOW-RESOURCE LANGUAGES

## ABSTRACT

Multilingual language models are widely used to extend NLP systems to low-resource languages. However, concrete evidence for the effects of multilinguality on language modeling performance in individual languages remains scarce. Here, we pre-train over 10,000 monolingual and multilingual language models for over 250 languages, including multiple language families that are understudied in NLP. We assess how language modeling performance in each language varies as a function of (1) monolingual dataset size, (2) added multilingual dataset size, (3) linguistic similarity of the added languages, and (4) model size (up to 45M parameters). We find that in moderation, adding multilingual data improves low-resource language modeling performance, similar to increasing low-resource dataset sizes by up to 33%. Improvements depend on the syntactic similarity of the added multilingual data, with marginal additional effects of vocabulary overlap. However, high-resource languages consistently perform worse in multilingual pre-training scenarios. As dataset sizes increase, adding multilingual data begins to hurt performance for both low-resource and high-resource languages, likely due to limited model capacity (the "curse of multilinguality"). These results suggest that massively multilingual pre-training may not be optimal for any languages involved, but that more targeted models can significantly improve performance.

## 1 INTRODUCTION

Multilingual language models have been a fixture of natural language processing (NLP) research nearly since the introduction of Transformer language models (Devlin et al., 2019; Conneau et al., 2020a). These models are often pre-trained on over 100 languages simultaneously, and they are widely used for NLP tasks in low-resource languages (Adelani et al., 2021; Ebrahimi et al., 2022; Hangya et al., 2022; Imani et al., 2023), cross-lingual transfer learning (Pires et al., 2019; Conneau et al., 2020a), and multilingual text generation (Lin et al., 2022; Scao et al., 2022). However, while multilingual language models produce strong results across many languages, multilingual pre-training work almost exclusively focuses on pre-training a small number of models with some fixed distribution over languages (e.g. mBERT, XLM-R, XGLM, and BLOOM; Devlin et al., 2019; Conneau et al., 2020a; Blevins et al., 2022; Lin et al., 2022; Scao et al., 2022).

Thus, it is largely unknown how different pre-training language distributions, such as different quantities of multilingual data or different selections of languages, affect multilingual language model performance. Multilingual models have been studied extensively during inference and fine-tuning (Pires et al., 2019; Conneau et al., 2020b; Karthikeyan et al., 2020; Winata et al., 2021; Chai et al., 2022; Alabi et al., 2022; Guarasci et al., 2022; Winata et al., 2022; Wu et al., 2022; Eronen et al., 2023), but these studies rely on the same sets of pre-trained models. Fujinuma et al. (2022) vary the set of pre-training languages, but they consider only 14 variations of 14 languages, and they focus on cross-lingual transfer after English fine-tuning. For within-language performance, there is mixed evidence for the benefits of multilingual vs. monolingual pre-training (Conneau et al., 2020a; Wu & Dredze, 2020; Pyysalo et al., 2021; §2). As multilingual language models are increasingly used without task-specific fine-tuning (e.g. for text generation; Scao et al., 2022; Lin et al., 2022), it is critical to better understand how multilingual pre-training affects raw language modeling performance in individual languages.

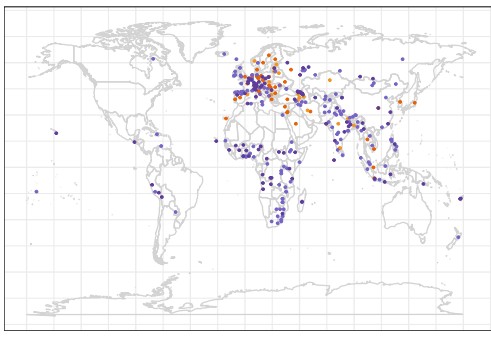 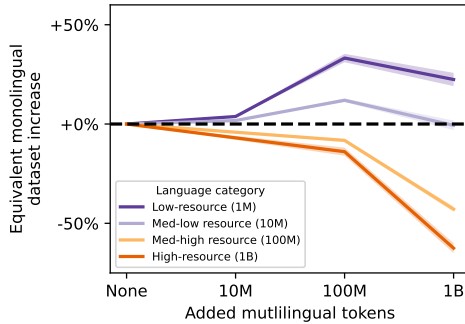

Figure 1: Left: Map of the 252 languages used in our study. Right: Effects of adding multilingual pre-training data in similar languages, for low-resource (1M token) through high-resource (1B token) languages in small models. Effects are quantified using the estimated monolingual dataset size that would achieve similar performance. Adding 1B tokens of multilingual data is similar to adding 22% (low-resource) or removing 63% (high-resource) of the monolingual dataset. Shaded regions are 99% confidence intervals for the mean.

In our work, we investigate the effects of different multilingual pre-training distributions on language modeling performance in 252 languages. Our main contributions are:[1]

- We pre-train over 1900 monolingual baseline models for 252 languages, and we estimate model performance in each language based on monolingual dataset size (§4). We use these estimates to quantify the performance of multilingual models in individual languages (§4.3).

- We pre-train over 8400 multilingual language models, and we evaluate how performance in individual languages varies as a function of monolingual dataset size, multilingual dataset size, linguistic similarity of the pre-training languages, and model size (up to 45M parameters; §5). By fixing monolingual tokenizers for all 252 languages, we are able to make valid perplexity comparisons even across multilingual models, and our results control for tokenization quality.

- We find that moderate amounts of multilingual data improve performance for low-resource languages, similar to increasing low-resource dataset sizes by up to 33% (§6.1). These improvements depend primarily on the syntactic similarity of the added multilingual data, with marginal additional effects of lexical (vocabulary) similarity.

- We find that multilingual data consistently hurts high-resource language performance, similar to reducing dataset sizes by over 85% in some cases (§6.2). Likely due to limited model capacity, as dataset sizes increase, adding multilingual data begins to hurt performance for both low-resource and high-resource languages (the *curse of multilinguality*; §2).

These results have significant practical implications for pre-training multilingual language models. The benefits of multilinguality on raw language modeling performance seem restricted to cases where both (1) the model targets performance in low-resource languages and (2) the model has enough capacity for the added multilingual data. If these assumptions hold, the multilingual data should be from languages that are linguistically similar to the target low-resource languages. However, when optimizing performance for multiple high-resource languages, multilingual models may quickly lead to intractable model sizes while degrading performance in individual languages.

## 2 RELATED WORK

**Multilingual language models for low-resource languages.** Recent work has adopted two primary strategies for extending language models to low-resource languages. The first is to pre-train one model on a large number of languages, including low-resource languages. This is the strategy adopted by models such as mBERT (104 languages; Devlin et al., 2019), XLM-R (100 languages; Conneau et al., 2020a), XGLM (30-100 languages; Lin et al., 2022), BLOOM (46 languages; Scao et al., 2022), and Glot500 (511 languages; Imani et al., 2023). Oftentimes, these models are later fine-tuned on a specific low-resource language (e.g. Ebrahimi et al., 2022). The second strategy

---

[1]Code will be available at `https://github.com/redacted-for-anonymity`.

is pre-train multilingual models on a smaller number of languages that are either closely related or spoken in a specific region. This strategy is adopted by models such as AfriBERTa (11 African languages; Ogueji et al., 2021) and IndicNLP (12 Indian languages; Kakwani et al., 2020).

The strategy of pre-training only on similar languages is based on evidence that cross-lingual transfer learning (e.g. fine-tuning on language $L_1$ and evaluating on $L_2$) occurs primarily between similar languages (Pires et al., 2019; Conneau et al., 2020b; Ahuja et al., 2022; Oladipo et al., 2022; Eronen et al., 2023). Features that have been proposed to drive cross-lingual transfer include the geographic proximity of languages (Winata et al., 2022), shared writing systems (Fujinuma et al., 2022; Imani et al., 2023), shared morphological systems (Gerz et al., 2018), and shared language families (Winata et al., 2022). However, Fujinuma et al. (2022) observe better cross-lingual transfer overall when a wider variety of languages is seen during during pre-training. In any case, these studies all focus on cross-lingual transfer during fine-tuning, rather than the effects of multilinguality on within-language performance or pre-training itself.

**The curse of multilinguality.** In fact, there is mixed evidence for whether multilingual pre-training improves downstream performance for individual languages. Conneau et al. (2020a) find that pre-training on an excessive number of languages hurts model performance in each language, evaluating five subsets of languages on downstream tasks in 16 languages. This phenomenon is known as the *curse of multilinguality* or *negative interference* (Wang et al., 2020). This result is further supported by findings that monolingual language models often have better language modeling performance than massively multilingual models such as mBERT (Pyysalo et al., 2021). However, Rust et al. (2021) find that this curse of multilinguality may simply be a result of lower quality tokenization per language in multilingual models. Furthermore, contradicting the curse of multilinguality, Wu & Dredze (2020) find that for low-resource languages, multilingual pre-training does improve downstream task performance relative to monolingual pre-training. Thus, the precise effects of multilinguality on low-resource and high-resource languages remain unclear.

To quantify these effects, we evaluate language modeling performance in 252 languages while systematically varying monolingual dataset size, multilingual dataset size, model size, and linguistic similarity of the added languages. This contrasts with previous studies that have focused only on individual multilingual models such as mBERT or XLM-R. Our approach allows us to determine how such models perform after varying pre-training languages and language distributions.

## 3    COLLECTING A MASSIVELY MULTILINGUAL DATASET

Conducting controlled multilingual language modeling experiments requires a large multilingual dataset. Notably, broad language coverage is a consistent issue in NLP research (Bender, 2009; 2011; Joshi et al., 2020; Blasi et al., 2022), and one contribution of our work is to compile references to text data sources for languages that are often under-studied in NLP.[2] We compile a dataset of text in 1572 languages; of these languages, 252 contain enough data (1.5M tokens) to be used in our language modeling study. While we are unable to redistribute our compiled dataset due to redistribution licenses and out of respect for the original data collectors, all of our sources are publicly available (§A.1). As a caveat, we note that many low-resource language datasets (e.g. language documentation projects) prohibit commercial use, and thus industry labs may be precluded from using such datasets without explicit permission from the owners.

We collect text corpora from 24 multilingual data sources such as OSCAR (Ortiz Suárez et al., 2019; Abadji et al., 2021), Wikipedia (Wikipedia, 2023), and No Language Left Behind (Costa-jussà et al., 2022). Our full list of sources and dataset collection details are reported in §A.1. We clean and concatenate the datasets for each language, and we deduplicate repeated sequences of 100 or more UTF-8 bytes (Lee et al., 2022). Restricting each language to a maximum of 1B tokens, our dataset contains 41.4B tokens in 1572 languages. This includes 1329 languages with at least 100K tokens (largely due to Bible translations) and 252 languages with the required 1.5M tokens for our language modeling study (1M tokens for pre-training and 500K tokens for evaluation). Despite this fairly stringent token requirement, our 252 languages cover five continents, 29 language families, and 30 scripts (i.e. writing systems). Figure 1 shows a geographic map of our 252 languages, using coordinates from Glottolog (Hammarström et al., 2023). Our list of languages is in §A.7.

---

[2]For other recent work on low-resource language dataset compilation, see Imani et al. (2023).

## 4 MONOLINGUAL BASELINES AND EVALUATION METRICS

To study effects of multilinguality on language modeling performance in individual languages, we first need a method to quantify performance in those languages. Thus, we pre-train 1989 monolingual baseline models for our 252 languages, to use as comparison points for the multilingual models in later sections. We consider three language model sizes and four dataset sizes per language when available. Then, we estimate the number of monolingual tokens in a language $L$ required to achieve a given level of performance in $L$. We use this estimated number of monolingual tokens as an interpretable performance metric for multilingual models.

### 4.1 MODEL ARCHITECTURES AND PRE-TRAINING

We pre-train autoregressive GPT-2 language models from scratch (Radford et al., 2019) with three sizes from Turc et al. (2019): tiny (4.6M parameters), mini (11.6M parameters), and small (29.5M parameters). For each language, we pre-train models with four dataset sizes when available: 1M, 10M, 100M, and 1B tokens, not including 500K tokens for evaluation in each case. We call these dataset sizes low, med-low, med-high, and high resource respectively. We have 252 languages with at least the low-resource dataset size, 167 with med-low resource, 48 with med-high resource, and 28 with high-resource. Our list of languages is in §A.7. Evaluation loss curves, model details, and full hyperparameters are reported in §A.3.

**Monolingual tokenizers.** We train a monolingual SentencePiece tokenizer with maximum vocabulary size 32K for each of our 252 languages (Kudo & Richardson, 2018), and we fix this tokenizer for all models pre-trained for that language. We train each tokenizer on 10K randomly-sampled lines of text in the language; for languages where more lines are available, the 10K-line tokenizers have reasonable vocabulary overlap with tokenizers trained on more lines (§A.2). For example, a 10K-line tokenizer on average covers 93.7% of the 4K most frequent tokens in the vocabulary of a 10M-line tokenizer. We restrict tokenizer training to 10K lines for all languages to control for tokenization quality across languages.

### 4.2 PERPLEXITY AND LOG-LIKELIHOOD EVALUATIONS

As an initial performance metric, we compute the log-likelihood assigned by a language model $\mathcal{M}$ to the unseen evaluation dataset for language $L$. Each of our monolingual models is evaluated on its corresponding pre-training language, but these methods also apply to our multilingual models (which each have a tokenizer fixed for one target language; §5). Averaging over tokens, evaluation log-likelihood is equivalent to negative log-perplexity, mean token log-probability, or the negative of the language model's cross-entropy loss (Equation 1). Because our tokenization remains fixed across all models with a given target language, perplexities and log-likelihoods are comparable within each target language. Higher log-likelihood scores indicate better language modeling performance, they are predictive of model performance on other natural language tasks (Xia et al., 2023), and they can be computed even for languages without any labeled datasets.

Although log-likelihood scores are comparable for models with the same target language, they vary substantially across languages. This can be due to features of individual languages, their datasets, or their tokenization (Gerz et al., 2018). Thus, when model $\mathcal{M}$ is pre-trained on language $L$, we subtract the log-likelihood score of the baseline tiny monolingual model (Baseline$_L$) trained on 1M tokens for that language, obtaining a relative log-likelihood as follows:

$$\text{Relative log-likelihood} = \text{mean}_w\big(\log_2 P_{\mathcal{M}}(w)\big) - \text{mean}_w\big(\log_2 P_{\text{Baseline}_L}(w)\big) \qquad (1)$$

Here, $w$ are tokens in the evaluation dataset for $L$. As is standard, token probabilities are produced by the language models $\mathcal{M}$ and Baseline$_L$ based on preceding context (Brown et al., 2020). Equation 1 is then equivalent to the log-odds of observing the evaluation dataset for $L$ using the model $\mathcal{M}$ versus the baseline model for $L$. Intuitively, a relative log-likelihood of $\ell$ in log base two indicates that $\mathcal{M}$ assigns the evaluation dataset $2^\ell$ times the likelihood assigned by the baseline model. Equivalently, $\mathcal{M}$ has perplexity $2^\ell$ times lower than the baseline model. In future sections, log-likelihoods refer to relative log-likelihoods that account for the target language baseline.

## 4.3 ESTIMATING MONOLINGUAL TOKEN COUNTS

However, relative log-likelihoods are difficult to interpret when quantifying language model performance in practice; a log-likelihood change of $1.0$ does not have concrete practical implications. Furthermore, log-likelihoods are difficult to compare across model sizes (§A.4). Therefore, when evaluating multilingual language models in later sections, we quantify performance in a language $L$ as the estimated number of monolingual tokens in $L$ that would achieve the same log-likelihood with the same size model. Measuring model performance in terms of estimated monolingual token counts allows us to quantify the effects of adding multilingual pre-training data across languages and model sizes.

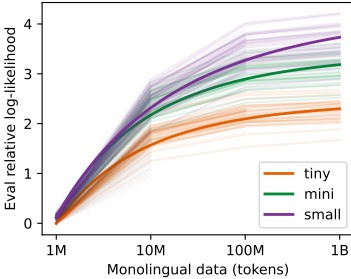 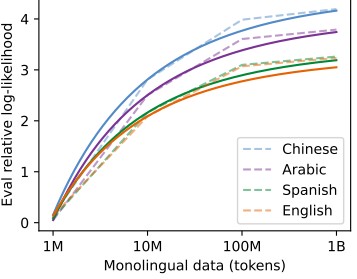

Figure 2: Curves predicting monolingual model performance from dataset size. Left: Curves fitted to all languages for each model size. Bold lines are fitted curves, and lighter lines are ground truth curves for individual languages. Right: Sample language-specific curves for small models, extrapolating from only two data points (1M and 10M tokens). This still produces reasonable estimates for 100M and 1B tokens. Bold lines are estimated curves, and dashed lines are ground truth values.

Estimating monolingual token counts for models across 252 languages is nontrivial. Previous work has found that language modeling loss (equivalent to negative log-likelihood) has a power law relationship with dataset size (Kaplan et al., 2020). Indeed, we find that $-ax^{-b} + c$ provides a good fit on average to relative log-likelihood in all 252 languages, where $x$ is the monolingual dataset size in $\log_{10}$ tokens (Figure 2, left). In line with previous work (Hoffmann et al., 2022), we observe that larger datasets improve performance primarily for larger models; at 1M tokens in any language, different model sizes perform similarly.

However, there is significant variability in the log-likelihood vs. dataset size curve across languages. For high-resource languages, we can fit a language-specific power law to the data points for 1M, 10M, 100M, and 1B tokens. For lower-resource languages, there are too few data points to fit the power law from scratch (e.g. three power law parameters with two data points). For these languages, we fix $a$ as the median parameter value from languages where the curve can be fit. Using this, we fit a monolingual log-likelihood vs. monolingual token count curve for each language in each model size (Figure 2, right; details in §A.4).

These curves produce reasonable estimates for the number of monolingual tokens required to achieve a given level of performance in a language $L$ (§A.4). Even when token estimation accuracy is imperfect, our estimated monolingual token count is always a monotonic increasing function of eval log-likelihood, and thus performance rankings between models are preserved. In future sections, we measure the performance of a multilingual model with target language $L$ in terms of the estimated number of monolingual pre-training tokens in $L$ that would achieve the same performance.

## 5 PRE-TRAINING MULTILINGUAL MODELS

Finally, we pre-train multilingual language models that vary along four dimensions: monolingual data quantity, added multilingual data quantity, model size, and linguistic similarity of the added languages. Each multilingual model is pre-trained with a specified target language, keeping monolingual tokenization for that language fixed during both pre-training and evaluation. The multilingual models are pre-trained identically to the monolingual baselines in §4, except adding one epoch of the multilingual data (i.e. 10M, 100M, or 1B tokens). The multilingual data is randomly interspersed with the monolingual pre-training data in the target language. Target language evaluation loss curves

are included in §A.3. In total, we pre-train 8454 multilingual language models ranging from 8M to 45M parameters.

**Multilingual tokenizers.** Perplexity and log-likelihood evaluations within a language $L$ are only comparable when they use the same tokenizer. Thus, we must keep the monolingual tokenizer fixed for any model evaluated on $L$. However, fixing tokenization for multiple languages simultaneously results in intractable vocabulary sizes. For example, 252 languages × 32K tokens would result in a vocabulary size of 8.1M tokens, requiring 1.0B embedding parameters even with our smallest embedding size of 128. To avoid intractable parameter counts, we pre-train multilingual language models that each keep tokenization fixed for only one language, which we call the *target language* for that model. In each multilingual model, the non-target languages share a multilingual tokenizer with vocabulary size 32K, trained on 10K randomly-sampled lines from each added language. The target language and added multilingual datasets are tokenized separately, and the token IDs are merged for the shared vocabulary items. This merged tokenization process ensures that the target language tokenization remains unchanged across models.

**Manipulated variables.** We manipulate four variables in our multilingual language models:

- **Monolingual data quantity.** As in §4, we consider four monolingual dataset sizes when available in the target language: 1M, 10M, 100M, and 1B tokens.

- **Multilingual data quantity.** We always add multilingual data from 10 languages, selected according to linguistic similarity as described below. We add an equal number of tokens from each language, totaling either 10M, 100M, or 1B tokens. To save pre-training computation resources, we omit the 10M added tokens scenario when the monolingual data is 100M or 1B tokens.

- **Linguistic similarity.** We use linguistic similarity to define which languages are added to the target language during multilingual pre-training. Due to limits on computational resources, we only consider two linguistic similarity levels: similar and dissimilar languages. Our linguistic similarity metric is based on three features: syntactic similarity, geographic proximity, and lexical similarity (i.e. tokenizer vocabulary overlap). Syntactic and geographic metrics are computed as cosine similarities between languages' syntactic and geographic vector representations from `lang2vec` (Littell et al., 2017), which pulls from the World Atlas of Language Structures (Dryer & Haspelmath, 2013). Lexical similarity is computed as the log number of shared tokens in the monolingual tokenizers for two languages (§4.1). We $Z$-score normalize each of these similarity metrics over all language pairs, and we define the linguistic similarity between any two languages as the mean of the three similarity scores. For example, the four most similar languages to English are Dutch, Swedish, Norwegian, and German. For each target language, we select either the ten most or least similar languages. To allow us to vary the multilingual data quantity without changing the added languages, we restrict our added languages to those with at least 100M tokens in our dataset (i.e. our 48 med-high resource languages).

- **Model size.** We use the same model sizes as §4. With the added multilingual vocabulary embeddings, the models have roughly 8.7M (tiny), 19.8M (mini), and 45.8M (small) parameters.

## 6 MULTILINGUAL MODEL RESULTS

We find that performance in low-resource languages improves when we add moderate amounts of multilingual data (§6.1). The amount of improvement depends on the syntactic similarity of the added languages, with small additional effects of lexical (vocabulary) similarity. High-resource language performance consistently degrades when we add multilingual data (§6.2). Larger models have smaller performance degradations for high-resource languages and larger performance improvements for low-resource languages in multilingual scenarios, suggesting that many drawbacks of multilinguality are due to limited model capacity.

### 6.1 LOW-RESOURCE LANGUAGE RESULTS

**In moderation, multilinguality improves low-resource performance.** As shown in Figure 3 (top), low-resource languages exhibit performance improvements when adding 100M or 1B tokens of multilingual data ($p < 0.001$ for 11 out of 12 comparisons, using paired sample $t$-tests; §A.5).

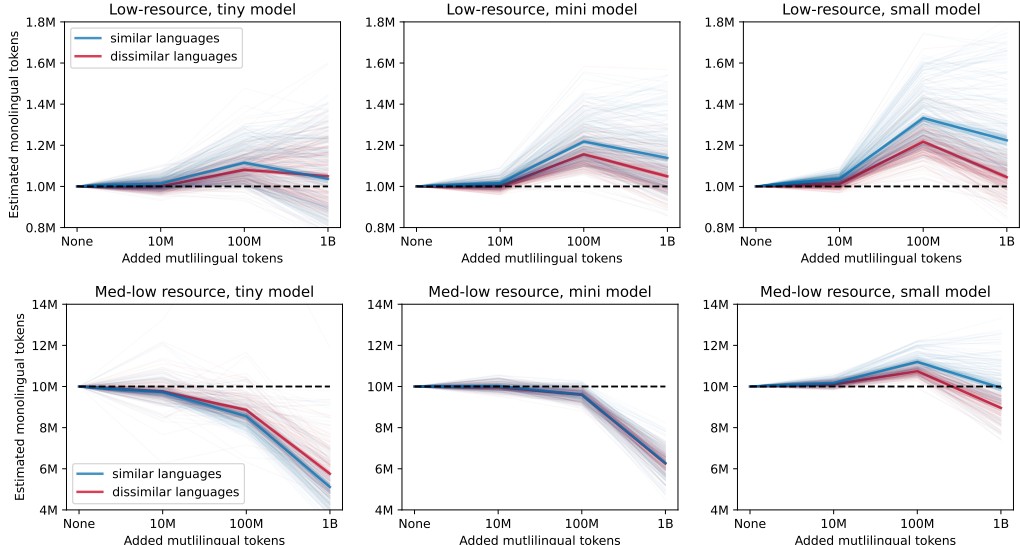

Figure 3: Results for low and med-low resource scenarios. Higher $y$-axis values indicate better performance. For example, a small model with 1M monolingual tokens (top right) and 1B added tokens of multilingual data in similar languages has similar performance to 1.2M monolingual tokens alone. Light-colored lines indicate results for individual languages, and bold lines indicate the mean across languages. Shaded regions are 95% confidence intervals for the mean.

Performance improvements are significantly larger when the added languages are similar vs. dissimilar to the target language (analogous to an average 33% vs. 22% increase in target language dataset size for small models in the optimal scenario; $p < 0.001$). Performance improvements are also larger for larger model sizes (33% vs. 12% equivalent dataset increases for small vs. tiny models; $p < 0.001$). Regardless of model size, performance is essentially unaffected when adding only 10M multilingual tokens (1M tokens in each added language); this result also holds for med-low resource scenarios (Figure 3, bottom). This suggests that a nontrivial amount of multilingual data is required for language models to leverage shared characteristics across languages.

However, the benefits of adding more multilingual data quickly plateau in low-resource scenarios (e.g. adding 100M vs. 1B multilingual tokens). In med-low resource scenarios (Figure 3, bottom), adding multilingual data hurts performance ($p < 0.001$ adding 1B multilingual tokens; §A.5) except in our largest models. Even in the larger models, the benefits of multilinguality decrease when too much multilingual data is added (Figure 3, right). This suggests that adding multilingual data is beneficial only in moderation, before models have reached their capacity limits.

**Syntactic similarity of added languages drives results.** We then investigate whether syntactic, geographic, or lexical (vocabulary) similarity of the added languages appears to drive multilingual model improvement. We focus on the low-resource small model scenario (Figure 3, top right) with 100M tokens of added multilingual data. This setup leads to our largest performance improvement on average for low-resource languages; other scenarios are considered in §A.6. We compute the mean syntactic, geographic, and lexical similarity of the added languages for each target language, both when selecting languages based on similarity and dissimilarity. All three similarity metrics correlate with model performance (relative log-likelihood scores), with Pearson's $r = 0.494$, $r = 0.341$, and $r = 0.346$ respectively (Figure 4, left, center). More similar added languages correlate with better performance. However, syntactic, geographic, and lexical similarity are also significantly correlated with one another ($r = 0.242$ to $0.602$). We use variance partitioning to determine the amount of variance in model performance accounted for by each feature, along with the variance accounted for by each feature after regressing out other features (Borcard et al., 1992; QCBS, 2023). We find that syntactic similarity of the added languages accounts for 24.2% of variance in multilingual model performance; adding geographic and lexical similarity increases this to only 26.4% (Figure 4, right). We note that syntactic similarity might reflect other typological features of languages or be serving as a proxy for taxonomic relatedness (Rama & Kolachina, 2012). Still, these

results suggest that abstract linguistic similarity drives the benefits of multilinguality more so than surface level features such as vocabulary overlap. This aligns with results for cross-lingual transfer during fine-tuning (Karthikeyan et al., 2020).

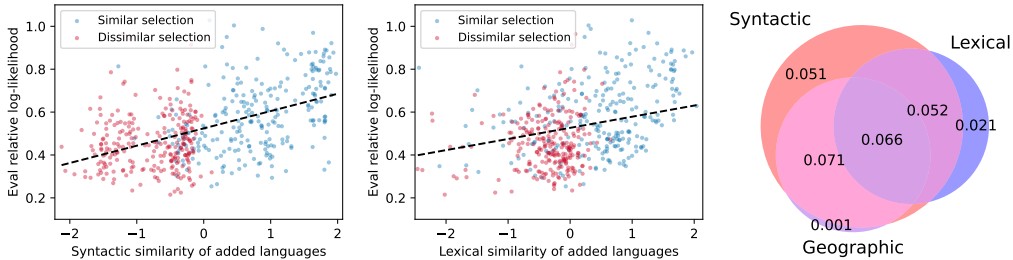

Figure 4: Left: Correlation between the mean syntactic similarity of the added languages and a model's relative log-likelihood score for the target language (Pearson's $r = 0.494$). Added languages are selected to be either similar or dissimilar (§5). A relative log-likelihood of $1.0$ indicates that the model assigns the eval dataset $2^{1.0}$ times the likelihood assigned by the baseline model for that language. Center: Correlation ($r = 0.346$) between the mean lexical (vocabulary) similarity of the added languages and a model's relative log-likelihood score. Right: Variance partitioning into syntactic, geographic, and lexical similarity of the added languages when predicting a model's relative log-likelihood score. Additional results in §A.6.

## 6.2 HIGH-RESOURCE LANGUAGE RESULTS

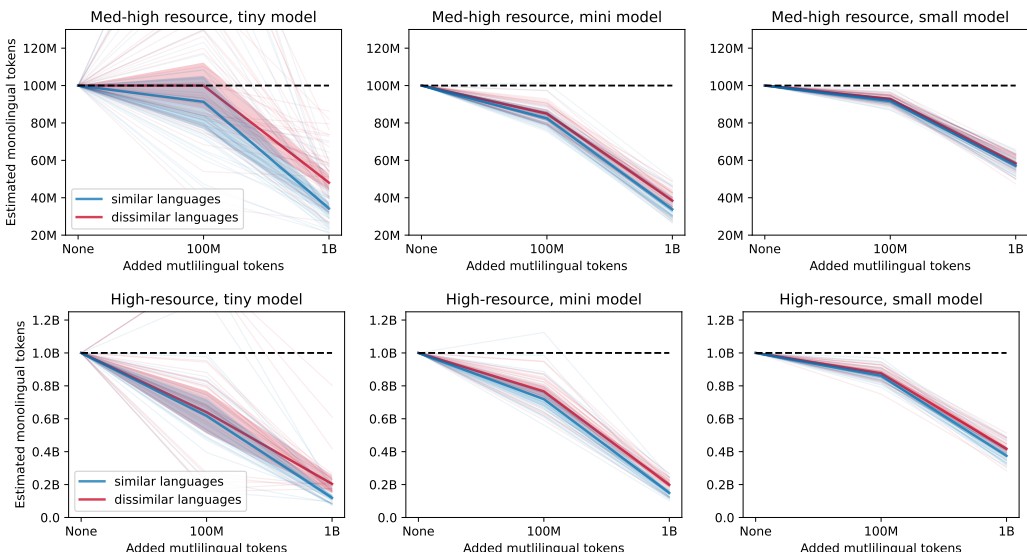

Figure 5: Results for med-high and high resource scenarios, using the same format as the low-resource scenarios in Figure 3. For example, adding 1B tokens of multilingual data to a small model with 1B monolingual tokens (high-resource; bottom right) is similar to removing over 600M tokens of the monolingual dataset.

**Multilinguality hurts high-resource performance.** For all model sizes, multilinguality hurts language model performance in med-high and high resource languages (Figure 5; $p < 0.001$ in all scenarios adding 1B tokens; §A.5). For high-resource languages in our largest model size, adding 1B multilingual tokens is similar to removing $63\%$ of the dataset in the target language. Degradations are larger when more multilingual tokens are added. Degradations are also larger for smaller models ($88\%$ vs. $63\%$ equivalent dataset decrease in the target language for tiny vs. small models; $p < 0.001$). This suggests that degradations due to multilinguality are likely driven by language models reaching their capacity limits. Interestingly, degradations are slightly larger given more similar added languages to the target language (all scenarios in Figure 5; $p < 0.05$ in 7 out of

12 scenarios). This indicates that although more similar languages tend to improve low-resource language performance (§6.1), they surprisingly tend to hurt high-resource language performance.

# 7 DISCUSSION

Our results demonstrate that for low-resource languages, multilingual language models yield some benefits. In the optimal case from our study, the benefits are similar to increasing the low-resource dataset size by about 33% (§6.1). Hence, in scenarios where collecting additional data is difficult (e.g. for languages spoken in remote geographic locations or with few speakers), pre-training multilingual models may be a worthwhile endeavor. In these cases, the models should be pre-trained with multilingual data from maximally similar languages, and it should be ensured that the models have capacity for the added multilingual data along with the target language data. However, in other cases, it may be more practical to simply find or collect more data in the target language itself.

For high-resource languages, multilingual language models yield worse performance than the comparable monolingual model in essentially all cases. Degradations can be similar to reducing high-resource dataset sizes by over 85% (§6.2). These degradations can be mitigated by pre-training larger models, which also appear to maximize benefits for low-resource languages. However, when pre-training language models even on the order of tens of high-resource languages (Conneau et al., 2020a; Scao et al., 2022; Lin et al., 2022), a model sufficiently large to accommodate all of the languages' data without hitting capacity limitations would be far too large to be practical. Even if existing large language models (LLMs) are severely over-parameterized, there is evidence that 70B-parameter models are required just for English (Hoffmann et al., 2022). If only considering performance in individual languages, pre-training targeted language-specific models is likely to be far more efficient than a single massively multilingual model.

## 7.1 LIMITATIONS

This work has several limitations. First, we only pre-train language models up to 45M parameters. Larger models are less likely to hit the capacity limitations that appear to drive the "curse of multilinguality". However, as discussed above, avoiding capacity limitations in multilingual models can quickly lead to intractable parameter counts. Particularly when pre-training thousands of models for controlled experiments, larger models may not be worth additional computational and environmental costs if results can reasonably be extrapolated to larger models (Strubell et al., 2019). In fact, for low-resource scenarios, smaller models can achieve similar performance to larger models (Figure 2) while remaining accessible to communities with fewer computational resources.

Second, while we have included more low-resource languages than the vast majority of recent studies in NLP, we do not have coverage of some regions and language families. For example, our study does not include any languages indigenous to modern-day Australia or many from the Americas. This imperfect coverage may lead our results to overestimate overall similarities between languages, and it may skew our results towards languages that have larger text corpora available on the Internet.

Finally, our results apply primarily to language modeling performance in individual languages. Effects of multilingual pre-training may be different for specific downstream tasks (e.g. reasoning tasks or machine translation; Bandarkar et al., 2023; Costa-jussà et al., 2022) or for cross-lingual transfer learning (Fujinuma et al., 2022). When pre-training multilingual language models, the specific downstream use cases for the models should be taken into consideration.

# 8 CONCLUSION

Our work systematically evaluates the effects of multilingual pre-training on language modeling performance in 252 languages. We pre-train over 10,000 monolingual and multilingual language models, varying monolingual dataset sizes, multilingual dataset sizes, linguistic similarity of the multilingual data, and model sizes. We find that adding multilingual data in similar languages improves performance for low-resource languages, but improvements decrease as models reach capacity limitations. Multilingual data consistently hurts high-resource language performance. This suggests that while multilingual language models may be beneficial for low-resource scenarios, massively multilingual models may be far less practical than previously assumed for raw language modeling.

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

# A APPENDIX

## A.1 DATASET DETAILS

We first download the first 32M lines for each language in the deduplicated September 2021 release of OSCAR (Ortiz Suárez et al., 2019; Abadji et al., 2021). We collect additional corpora

for languages with less than 1M lines in OSCAR (approximately 50M tokens, based on OS-CAR line lengths) and for languages that do not appear in OSCAR. Additional corpora include: Wikipedia (Wikipedia, 2023), No Language Left Behind (Costa-jussà et al., 2022), the Leipzig Corpora Collection (Goldhahn et al., 2012), eBible translations (eBible, 2023), FLORES-200 (Costa-jussà et al., 2022), Tatoeba (Tiedemann, 2012; 2020), AfriBERTa (Ogueji et al., 2021), NusaX (Winata et al., 2023), AmericasNLP (Mager et al., 2021), AmericasNLI (Ebrahimi et al., 2022), the Nunavut Hansard Inuktitut–English Parallel Corpus (Joanis et al., 2020), the Cherokee-English ChrEn dataset (Zhang et al., 2020), the Cherokee Corpus (Cherokee Corpus, 2023), the Cree Corpus (Teodorescu et al., 2022), Languages of Russia (Zaydelman et al., 2016), the Evenki Life newspaper (Zueva et al., 2020), the transcribed Fula Speech Corpora (Cawoylel, 2023), IsiXhosa (Podile & Eiselen, 2016), the Ewe Language Corpus (Gbedevi Akouyo et al., 2021), the Makerere Luganda Corpora (Mukiibi et al., 2022), the CMU Haitian Creole dataset (CMU, 2010), the Tigrinya Language Modeling Dataset (Gaim et al., 2021), and Ulukau (Ulukau, 2023). Our Wikipedia corpora use the Wikimedia dump from August 20, 2023 (Wikimedia, 2023). All other corpora use their publicly available versions as of August 2023. Links to individual corpora are included at https://github.com/redacted-for-anonymity.

We clean these corpora by removing lines containing only repetitive characters, exact duplicate lines, and lines identified as English by the spaCy language detection tool with confidence above 0.95 (Honnibal et al., 2020). We find that English filtering is particularly important for Wikipedia, from which we also remove redundant lists of links and headers. We manually inspect all files for egregious unclean text lines, and we remove any patterns found.

All corpora outside of OSCAR are truncated to 2M cleaned lines per language, which encompasses the entire corpus for most datasets; for example, only 4 out of 239 downloaded Wikipedias are truncated (recall that we only download additional corpora for languages with less than 1M lines in OSCAR). After merging corpora per language, repeated sequences of 100 UTF-8 bytes are deduplicated using the code from Lee et al. (2022). Corpora are unshuffled unless their public release is already shuffled. This allows tokenized sequences to span multiple consecutive lines; the tokenized sequences are shuffled prior to language model pre-training. Final token counts per language are listed in §A.7.

## A.2 Tokenization Quality

To control for tokenization quality across languages, all of our monolingual tokenizers are SentencePiece tokenizers trained on 10K lines of text with maximum vocabulary size 32K (§4.1; Kudo & Richardson, 2018). We have at least 10K lines of text in each of our 252 languages. All evaluations (including for multilingual models, which fix the target language monolingual tokenizer) are conducted using these tokenizers. The multilingual tokenizers in §5 are used only for added data during multilingual pre-training; they are not used for evaluation. To ensure that our monolingual tokenizers have reasonable quality, we compare their vocabularies with tokenizers trained on more lines of text. Specifically, for each of our 28 high-resource languages, we train tokenizers on 10K, 100K, 1M, and 10M lines of text. For each training dataset size, we compute the vocabulary overlap with the 4K and 8K most frequent tokens in the 10M-line tokenizer (the "reference vocabulary"). Figure 6 shows the reference vocabulary overlap for the different training dataset sizes. At 10K lines, the tokenizer vocabularies on average cover $93.7\%$ of the 4K-token reference vocabulary and $87.8\%$ of the 8K-token reference vocabulary, indicating reasonable tokenization quality.

## A.3 Language Model Pre-Training Details

Language models are pre-trained using the Hugging Face Transformers library (Wolf et al., 2020) and code from Chang & Bergen (2022). Hyperparameters are reported in Table 1 (left). All of our models use the GPT-2 architecture (Radford et al., 2019), changing only the number of layers, attention heads, and embedding sizes as in Turc et al. (2019). Models are pre-trained for 20 epochs of the target language monolingual data in the low and med-low resource scenarios, 10 epochs in the med-high resource scenario, and 2 epochs in the high-resource scenario. Based on initial results using randomly-sampled languages, pre-training on more than 20 epochs often leads to overfitting (increases in eval loss) in low-resource scenarios. Multilingual models include one epoch of the multilingual data (§5) randomly interspersed with the target language data. The numbers of pre-

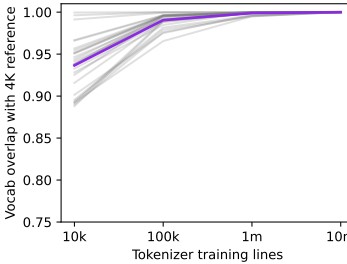 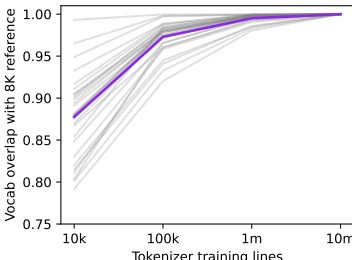

Figure 6: Vocabulary overlap with the reference vocabulary for tokenizers trained on different numbers of lines. The reference vocabulary consists of the 4K (left) or 8K (right) most frequent tokens in a 10M-line tokenizer for that language. We report the percentage of the reference vocabulary that is covered by 32K-vocabulary tokenizers with different training dataset sizes. Gray lines indicate individual languages, and the purple line indicates the mean across languages.

training steps for different dataset configurations are reported in Table 1 (right). Average evaluation loss curves during pre-training are shown in Figure 7. For each target language, the same 500K evaluation tokens are held out in all cases. In the monolingual low-resource scenario for each language (i.e. 1M pre-training tokens), we pre-train three tiny models (instead of one) and compute their average evaluation log-likelihood, because these models are used as the baseline models for relative log-likelihoods (§4.2).

All language model pre-training runs together take a total of $1.87 \times 10^{20}$ FLOPs. This is less than $1/1500\times$ the computation used to train the original 175B-parameter GPT-3 model (Brown et al., 2020; $3.14 \times 10^{23}$ FLOPs). Models are each trained on one NVIDIA GeForce GTX TITAN X, GeForce RTX 2080 Ti, TITAN Xp, Quadro P6000, RTX A4500, RTX A5000, or RTX A6000 GPU. Our pre-training experiments take approximately 17700 A6000 GPU hours. Dataset cleaning, tokenization, and merging takes approximately 5880 CPU core hours, largely due to dataset tokenization with each multilingual tokenizer.

| Hyperparameter | Tiny | Mini | Small |
|---|---|---|---|
| Layers | 2 | 4 | 4 |
| Embedding size | 128 | 256 | 512 |
| Hidden size | 128 | 256 | 512 |
| Intermediate hidden size | 512 | 1024 | 2048 |
| Attention heads | 2 | 4 | 8 |
| Attention head size | 64 | 64 | 64 |
| Learning rate | 1e-3 | 7e-4 | 5e-4 |
| Activation function | | | GELU |
| Max sequence length | | | 128 |
| Position embedding | | | Absolute |
| Batch size | | | 128 |
| Learning rate decay | | | Linear |
| Warmup steps | | 10% of pre-training | |
| Adam $\epsilon$ | | | 1e-6 |
| Adam $\beta_1$ | | | 0.9 |
| Adam $\beta_2$ | | | 0.999 |
| Dropout | | | 0.1 |
| Attention dropout | | | 0.1 |

| Mono. tokens | Mono. epochs | Multi. tokens | Pre-training steps |
|---|---|---|---|
| 1M | 20 | 0 | 1250 |
| 1M | 20 | 10M | 1875 |
| 1M | 20 | 100M | 7500 |
| 1M | 20 | 1B | 63750 |
| 10M | 20 | 0 | 12500 |
| 10M | 20 | 10M | 13125 |
| 10M | 20 | 100M | 18750 |
| 10M | 20 | 1B | 75000 |
| 100M | 10 | 0 | 62500 |
| 100M | 10 | 100M | 68750 |
| 100M | 10 | 1B | 125000 |
| 1B | 2 | 0 | 125000 |
| 1B | 2 | 100M | 131250 |
| 1B | 2 | 1B | 187500 |

Table 1: Left: Language model pre-training hyperparameters (Devlin et al., 2019; Turc et al., 2019; Radford et al., 2018). To prevent overfitting (increasing loss on the eval dataset), learning rates are halved for mini and small models in the low-resource scenario, to 4e-4 and 2e-4 respectively (§4.1). Right: Pre-training steps for different monolingual and multilingual dataset sizes. There is always one epoch of the multilingual dataset (§5).

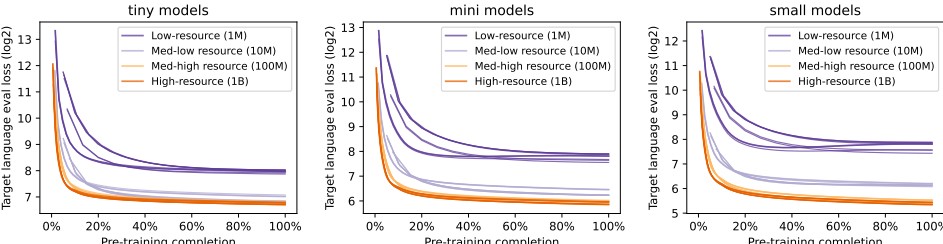

Figure 7: Target language evaluation loss curves during pre-training, for different model sizes and language resource scenarios. Each individual curve corresponds to a dataset configuration in Table 1 (right), averaging the loss curve over languages.

### A.4 MONOLINGUAL TOKEN ESTIMATION DETAILS

We overview our monolingual token estimation process in §4.3, and we provide details here. As motivation, we note that relative log-likelihood scores are not comparable across model sizes. For example, suppose that adding a multilingual dataset $D$ improves a model's eval log-likelihood score by $1.0$ in both small and large models. In this case, it would be unclear whether the effect of $D$ is intuitively "equal" in the two model sizes; doubling the likelihood of the eval dataset is likely more difficult in the larger model, so we might interpret $D$ as having a larger effect on the larger model despite the same change in log-likelihood. To avoid this ambiguity, we measure model performance using the estimated number of monolingual tokens in the target language that would achieve similar performance. In the case above, adding the multilingual dataset $D$ might be similar to adding $n_1$ monolingual tokens to the smaller model, but similar to adding $n_2 > n_1$ monolingual tokens to the larger model.

To estimate this, we first fit a power law $-ax^{-b} + c$ for each of our 252 languages, predicting a model's relative log-likelihood score (§4.2) based on its pre-training dataset size in log10 tokens. Each language has up to four ground truth values, corresponding to our monolingual models pre-trained on 1M, 10M, 100M, and 1B tokens. When all four points are available (i.e. our 28 high-resource languages), we are able to fit a power law from scratch. From these languages, we estimate the medians and standard deviations of $a$, $b$, and $c$. For languages with fewer than four data points, we constrain $a$, $b$, and $c$ to be within $2.5$ standard deviations from the median parameter value. If this leads the curve fitting to diverge, we loosen this constraint to $5.0$, $7.5$, then $10.0$ standard deviations from the median.

For languages where the curve fitting still does not converge or languages with too few data points (e.g. med-low resource languages with data points only for 1M and 10M tokens), we fix $a$ as the median parameter value from the high-resource languages. We fit only $b$ and $c$, which we constrain using standard deviations in the same way as described above. If the curve fitting still does not converge when fixing $a$ (e.g. low-resource languages with a data point only for 1M tokens), we fix both $a$ and $b$ as their median values. In that case, we only fit $c$, which is equivalent to simply shifting the median curve up or down by a constant. All curve fitting is implemented using `scipy` (Virtanen et al., 2020).

Finally, in many cases, we compare multilingual models to monolingual models with a specific known dataset size. The multilingual models in §6 are all compared to corresponding monolingual models without any added multilingual data. For example, a multilingual model with 10M monolingual tokens and 100M added multilingual tokens (relative log-likelihood score $y_1$) would be compared to a monolingual model with 10M monolingual tokens alone (relative log-likelihood score $y_0$). In these cases, we constrain our curve-fitting to pass through the point corresponding to the reference monolingual model (e.g. in the example described, the curve would be required to pass through the ground truth point $(7.0, y_0)$ for $10^{7.0}$ monolingual tokens alone). This only slightly alters the curve predicting relative log-likelihood score from log10 tokens, but it ensures that our baseline monolingual models in §6 lie exactly at 1M, 10M, 100M, and 1B tokens (Figures 3 and 5).

Once we have fitted a curve predicting a model's relative log-likelihood score from log10 pre-training tokens in a language $L$, we use this curve to estimate the number of tokens required to

achieve any relative log-likelihood score. Then, we have two metrics for a multilingual model's performance on target language $L$: (1) the model's relative log-likelihood score itself and (2) the estimated number of monolingual tokens in $L$ that would achieve that relative log-likelihood. The latter metric is easily interpretable, and it facilitates comparisons across languages and model sizes. We note that the estimated token count is a monotonic increasing function of relative log-likelihood score in all cases. Thus, even if the estimated token counts are not perfectly accurate, they preserve performance rankings between models (e.g. between our multilingual models and the monolingual baselines). A language model with target language $L$ will have a higher estimated token count if and only if it assigns a higher log-likelihood score to the evaluation dataset for $L$.

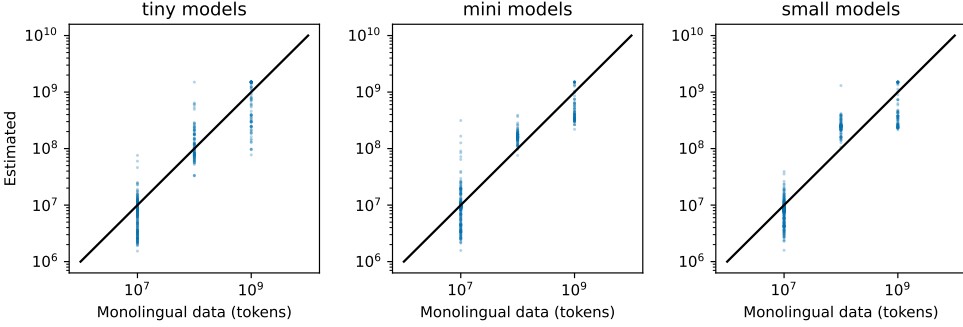

Figure 8: Estimated monolingual token counts for held-out monolingual models. Token counts are estimated from each model's relative log-likelihood score using a curve fitted to the specific language (§A.4). Estimations are extrapolating one order of magnitude out from the points used to fit the curve. In practice, we generally do not need to extrapolate this far for our results. The black line indicates perfect accuracy.

Still, we evaluate the quality of our monolingual token count estimation process. For each language $L$, we have up to four monolingual models (1M, 10M, 100M, and 1B pre-training tokens). We hold out one (or multiple) of the models, and we estimate its monolingual token count based on a curve fitted to the other monolingual models for $L$. We note that these estimations are extrapolating at minimum one order of magnitude away from the models used to fit the curve, because the models are exactly one order of magnitude apart in terms of pre-training tokens. The results in §6 do not need to extrapolate this far. Still, even with this larger extrapolation, we obtain reasonable estimates of monolingual token counts in the held-out scenarios (Figure 8). The root-mean-square errors are 0.340, 0.317, and 0.335 log10 tokens for tiny, mini, and small models respectively.

### A.5 STATISTICAL TESTS

We run paired sample $t$-tests to assess the statistical significance of our results from §6. For each reported $p$-value, we compare models that differ by exactly one of: monolingual dataset size, multilingual dataset size, linguistic similarity of the added languages, or model size. We pair models by language, so each pair differs by only the manipulated variable. To avoid potential artifacts from our token estimation process, we compare model relative log-likelihoods directly (§4.2) unless comparing across two model sizes (because relative log-likelihood improvements and degradations are difficult to compare across model sizes; §A.4). If comparing across model sizes, we compare the estimated monolingual token counts of the models. In both cases, we use a paired sample $t$-test. To decrease the chance of false positive results, we only run the statistical tests whose $p$-values are reported in the main text, and we account for multiple comparisons using Bonferroni correction (Bonferroni, 1936). For estimates of significance, the plots in §6 also include 95% confidence intervals for means.

### A.6 EFFECTS OF LINGUISTIC SIMILARITY ON MODEL PERFORMANCE

In §6.1, we find that the mean syntactic similarity of the added languages accounts for more variance in multilingual model performance (relative log-likelihood scores) than geographic and lexical (vocabulary) similarity. In that section, we consider the low-resource scenario with 100M added

multilingual tokens in small models. Here, we report the same results for tiny, mini, and small models. Variance partitioning results are shown in Figure 9. In all cases, syntactic similarity accounts for more variance than geographic and lexical similarity. Correlations between different similarity measures and model performance for mini models with 100M added multilingual tokens are plotted in Figure 10.

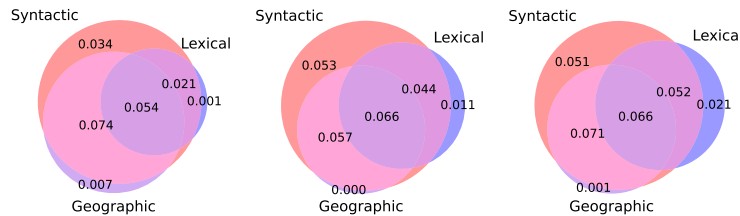

Figure 9: Variance partitioning into syntactic, geographic, and lexical similarity of the added languages when predicting a model's performance (relative log-likelihood score) for tiny (left), mini (center), and small (right) models with 100M tokens of added multilingual data.

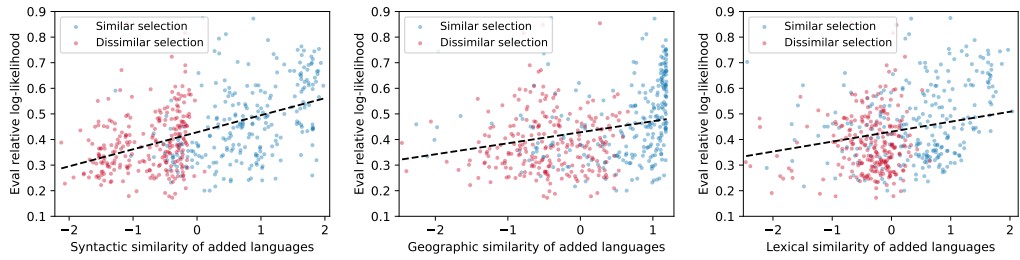

Figure 10: Correlations between different similarity measures (between target language and added languages) and multilingual model performance (relative log-likelihood) in the target language.

### A.7 LIST OF LANGUAGES

The 252 languages included in our language modeling study are listed in Table 2. These languages are those with at least 1.5M tokens in our dataset (§A.1). We restrict all languages to a maximum of 1B tokens. In lower resource scenarios, higher resource languages are subsampled to mimic the lower resource scenario. For example, we have 167 med-low resource languages when including the subsampled med-high and high resource languages. We distinguish between the same language in multiple scripts (e.g. Serbian in Cyrillic vs. Latin script) and macrolanguages vs. their individual constituent languages (e.g. Quechua vs. Cusco Quechua and Ayacucho Quechua). The full list of 1572 languages in our dataset can be found at `https://github.com/redacted-for-anonymity`.

| | Language | Language (ISO 639-3) | Script (ISO 15924) | Tokens | Resource Category | Language Family |
|---|---|---|---|---|---|---|
| 1 | Bulgarian | bul | cyrl | 1024512000 | high | Indo-European |
| 2 | Chinese | zho | hans | 1024512000 | high | Sino-Tibetan |
| 3 | Czech | ces | latn | 1024512000 | high | Indo-European |
| 4 | Danish | dan | latn | 1024512000 | high | Indo-European |
| 5 | Dutch | nld | latn | 1024512000 | high | Indo-European |
| 6 | English | eng | latn | 1024512000 | high | Indo-European |
| 7 | Finnish | fin | latn | 1024512000 | high | Uralic |
| 8 | French | fra | latn | 1024512000 | high | Indo-European |
| 9 | German | deu | latn | 1024512000 | high | Indo-European |
| 10 | Hebrew | heb | hebr | 1024512000 | high | Afro-Asiatic |
| 11 | Hungarian | hun | latn | 1024512000 | high | Uralic |
| 12 | Indonesian | ind | latn | 1024512000 | high | Austronesian |
| 13 | Iranian Persian | pes | arab | 1024512000 | high | Indo-European |

| 14 | Italian | ita | latn | 1024512000 | high | Indo-European |
|----|---------|-----|------|-----------|------|---------------|
| 15 | Japanese | jpn | jpan | 1024512000 | high | Japonic |
| 16 | Korean | kor | hang | 1024512000 | high | Koreanic |
| 17 | Modern Greek | ell | grek | 1024512000 | high | Indo-European |
| 18 | Polish | pol | latn | 1024512000 | high | Indo-European |
| 19 | Portuguese | por | latn | 1024512000 | high | Indo-European |
| 20 | Romanian | ron | latn | 1024512000 | high | Indo-European |
| 21 | Russian | rus | cyrl | 1024512000 | high | Indo-European |
| 22 | Spanish | spa | latn | 1024512000 | high | Indo-European |
| 23 | Standard Arabic | arb | arab | 1024512000 | high | Afro-Asiatic |
| 24 | Swedish | swe | latn | 1024512000 | high | Indo-European |
| 25 | Thai | tha | thai | 1024512000 | high | Tai-Kadai |
| 26 | Turkish | tur | latn | 1024512000 | high | Turkic |
| 27 | Ukrainian | ukr | cyrl | 1024512000 | high | Indo-European |
| 28 | Vietnamese | vie | latn | 1024512000 | high | Austro-Asiatic |
| 29 | Lithuanian | lit | latn | 787855616 | medhigh | Indo-European |
| 30 | Hindi | hin | deva | 774095488 | medhigh | Indo-European |
| 31 | Catalan | cat | latn | 771223680 | medhigh | Indo-European |
| 32 | Slovak | slk | latn | 746472192 | medhigh | Indo-European |
| 33 | Norwegian Bokmål | nob | latn | 612469888 | medhigh | Indo-European |
| 34 | Estonian | est | latn | 500367232 | medhigh | Uralic |
| 35 | Bengali | ben | beng | 419860608 | medhigh | Indo-European |
| 36 | Latvian | lav | latn | 379466368 | medhigh | Indo-European |
| 37 | Serbian | srp | cyrl | 279173376 | medhigh | Indo-European |
| 38 | Slovenian | slv | latn | 270027392 | medhigh | Indo-European |
| 39 | Tamil | tam | taml | 257684608 | medhigh | Dravidian |
| 40 | Albanian | sqi | latn | 240805504 | medhigh | Indo-European |
| 41 | Azerbaijani | aze | latn | 178155008 | medhigh | Turkic |
| 42 | Urdu | urd | arab | 143181312 | medhigh | Indo-European |
| 43 | Nepali | npi | deva | 139989120 | medhigh | Indo-European |
| 46 | Macedonian | mkd | cyrl | 124803328 | medhigh | Indo-European |
| 47 | Kazakh | kaz | cyrl | 124020480 | medhigh | Turkic |
| 48 | Georgian | kat | geor | 122249472 | medhigh | Kartvelian |
| 49 | Armenian | hye | armn | 121111040 | medhigh | Indo-European |
| 50 | Belarusian | bel | cyrl | 108812544 | medhigh | Indo-European |
| 44 | Esperanto | epo | latn | 102911872 | medlow | Constructed |
| 45 | Croatian | hrv | latn | 102911872 | medlow | Indo-European |
| 51 | Malayalam | mal | mlym | 90062848 | medlow | Dravidian |
| 52 | Icelandic | isl | latn | 88493056 | medlow | Indo-European |
| 53 | Welsh | cym | latn | 86114176 | medlow | Indo-European |
| 54 | Telugu | tel | telu | 81769088 | medlow | Dravidian |
| 55 | Galician | glg | latn | 81455616 | medlow | Indo-European |
| 56 | Hausa | hau | latn | 81195520 | medlow | Afro-Asiatic |
| 57 | Mongolian | mon | cyrl | 79270528 | medlow | Mongolic |
| 58 | Marathi | mar | deva | 78900992 | medlow | Indo-European |
| 59 | Asturian | ast | latn | 76998272 | medlow | Indo-European |
| 60 | Afrikaans | afr | latn | 75925632 | medlow | Indo-European |
| 61 | Basque | eus | latn | 75490304 | medlow | Basque |
| 62 | Burmese | mya | mymr | 75295104 | medlow | Sino-Tibetan |
| 63 | Bosnian | bos | latn | 73321472 | medlow | Indo-European |
| 64 | Central Kanuri | knc | arab | 72147840 | medlow | Nilo-Saharan |
| 65 | Somali | som | latn | 71963648 | medlow | Afro-Asiatic |
| 66 | Tatar | tat | cyrl | 71448448 | medlow | Turkic |
| 67 | Cebuano | ceb | latn | 71133568 | medlow | Austronesian |
| 68 | Kannada | kan | knda | 69977600 | medlow | Dravidian |
| 69 | Central Khmer | khm | khmr | 67915392 | medlow | Austro-Asiatic |
| 70 | Gujarati | guj | gujr | 65388416 | medlow | Indo-European |
| 71 | Panjabi | pan | guru | 64354560 | medlow | Indo-European |
| 72 | Bashkir | bak | cyrl | 64024832 | medlow | Turkic |
| 73 | Central Kurdish | ckb | arab | 60765440 | medlow | Indo-European |
| 74 | Maltese | mlt | latn | 59164544 | medlow | Afro-Asiatic |

| 75 | Serbo-Croatian | hbs | latn | 58518784 | medlow | Indo-European |
|---|---|---|---|---|---|---|
| 76 | Tajik | tgk | cyrl | 57351424 | medlow | Indo-European |
| 77 | Tagalog | tgl | latn | 55507456 | medlow | Austronesian |
| 78 | Kirghiz | kir | cyrl | 55496576 | medlow | Turkic |
| 79 | Tigrinya | tir | ethi | 55378816 | medlow | Afro-Asiatic |
| 80 | Malay | msa | latn | 55249152 | medlow | Austronesian |
| 81 | Igbo | ibo | latn | 53409920 | medlow | Niger-Congo |
| 82 | Sinhala | sin | sinh | 53101952 | medlow | Indo-European |
| 83 | Irish | gle | latn | 51020544 | medlow | Indo-European |
| 84 | Amharic | amh | ethi | 49825536 | medlow | Afro-Asiatic |
| 85 | Uzbek | uzb | latn | 49750144 | medlow | Turkic |
| 86 | Swahili | swa | latn | 49580928 | medlow | Atlantic-Congo |
| 87 | Luxembourgish | ltz | latn | 46355968 | medlow | Indo-European |
| 88 | Yoruba | yor | latn | 45996544 | medlow | Niger-Congo |
| 89 | Haitian | hat | latn | 43803264 | medlow | Creole |
| 90 | Kinyarwanda | kin | latn | 42016128 | medlow | Niger-Congo |
| 91 | Samoan | smo | latn | 41137664 | medlow | Austronesian |
| 92 | Javanese | jav | latn | 40730368 | medlow | Austronesian |
| 93 | Norwegian Nynorsk | nno | latn | 40680192 | medlow | Indo-European |
| 94 | Lao | lao | laoo | 40182528 | medlow | Tai-Kadai |
| 95 | Nyanja | nya | latn | 39635968 | medlow | Niger-Congo |
| 96 | Sindhi | snd | arab | 39586304 | medlow | Indo-European |
| 97 | Southern Pashto | pbt | arab | 39270656 | medlow | Indo-European |
| 98 | Sundanese | sun | latn | 39227648 | medlow | Austronesian |
| 99 | Maori | mri | latn | 39110528 | medlow | Austronesian |
| 100 | Occitan | oci | latn | 39094784 | medlow | Indo-European |
| 101 | Plateau Malagasy | plt | latn | 38467200 | medlow | Austronesian |
| 102 | Pushto | pus | arab | 37981184 | medlow | Indo-European |
| 103 | Scottish Gaelic | gla | latn | 37471488 | medlow | Indo-European |
| 104 | Shona | sna | latn | 37057152 | medlow | Niger-Congo |
| 105 | Waray | war | latn | 36727424 | medlow | Austronesian |
| 106 | Zulu | zul | latn | 36472960 | medlow | Niger-Congo |
| 107 | Dari | prs | arab | 36289920 | medlow | Indo-European |
| 108 | Northern Uzbek | uzn | latn | 35988736 | medlow | Turkic |
| 109 | Uighur | uig | arab | 35028992 | medlow | Turkic |
| 110 | Assamese | asm | beng | 34396032 | medlow | Indo-European |
| 111 | Southern Sotho | sot | latn | 34028544 | medlow | Niger-Congo |
| 112 | Lushai | lus | latn | 33796480 | medlow | Sino-Tibetan |
| 113 | Standard Malay | zsm | latn | 32638592 | medlow | Austronesian |
| 114 | Xhosa | xho | latn | 31847680 | medlow | Niger-Congo |
| 115 | Sicilian | scn | latn | 31407104 | medlow | Indo-European |
| 116 | Lombard | lmo | latn | 31299456 | medlow | Indo-European |
| 117 | Eastern Yiddish | ydd | hebr | 30456448 | medlow | Indo-European |
| 118 | Egyptian Arabic | arz | arab | 30198528 | medlow | Afro-Asiatic |
| 119 | Limburgan | lim | latn | 30182912 | medlow | Indo-European |
| 120 | Odia | ory | orya | 29186688 | medlow | Indo-European |
| 121 | South Azerbaijani | azb | arab | 29091584 | medlow | Turkic |
| 122 | Ayacucho Quechua | quy | latn | 29080448 | medlow | Quechuan |
| 123 | West Central Oromo | gaz | latn | 27978240 | medlow | Afro-Asiatic |
| 124 | Halh Mongolian | khk | cyrl | 27626624 | medlow | Mongolic |
| 125 | Venetian | vec | latn | 26978816 | medlow | Indo-European |
| 126 | Banjar | bjn | latn | 26552448 | medlow | Austronesian |
| 127 | Gilaki | glk | arab | 26084736 | medlow | Indo-European |
| 128 | Ganda | lug | latn | 25706752 | medlow | Niger-Congo |
| 129 | Papiamento | pap | latn | 24957568 | medlow | Creole |
| 130 | Sanskrit | san | deva | 24549760 | medlow | Indo-European |
| 131 | Rundi | run | latn | 24451072 | medlow | Niger-Congo |
| 132 | Chinese | zho | hant | 23736832 | medlow | Sino-Tibetan |
| 133 | Achinese | ace | latn | 23719936 | medlow | Austronesian |
| 134 | Tswana | tsn | latn | 23584384 | medlow | Niger-Congo |
| 135 | Western Panjabi | pnb | arab | 22000640 | medlow | Indo-European |

| 136 | Twi | twi | latn | 21262208 | medlow | Atlantic-Congo |
|-----|-----|-----|------|----------|--------|----------------|
| 137 | Iloko | ilo | latn | 21032576 | medlow | Austronesian |
| 138 | Chechen | che | cyrl | 20793856 | medlow | Nakh-Daghestanian |
| 139 | Tsonga | tso | latn | 20281984 | medlow | Niger-Congo |
| 140 | Yakut | sah | cyrl | 19829248 | medlow | Turkic |
| 141 | Western Frisian | fry | latn | 19808384 | medlow | Indo-European |
| 142 | Kurdish | kur | latn | 19233152 | medlow | Indo-European |
| 143 | Ewe | ewe | latn | 18750848 | medlow | Niger-Congo |
| 144 | Oriya | ori | orya | 18473216 | medlow | Indo-European |
| 145 | Latin | lat | latn | 17430272 | medlow | Indo-European |
| 146 | Chuvash | chv | cyrl | 16924288 | medlow | Turkic |
| 147 | Minangkabau | min | latn | 16113024 | medlow | Austronesian |
| 148 | Faroese | fao | latn | 15750272 | medlow | Indo-European |
| 149 | Breton | bre | latn | 14796032 | medlow | Indo-European |
| 150 | Yue Chinese | yue | hant | 14777472 | medlow | Sino-Tibetan |
| 151 | Pedi | nso | latn | 14619264 | medlow | Niger-Congo |
| 152 | Tosk Albanian | als | latn | 14432000 | medlow | Indo-European |
| 153 | Crimean Tatar | crh | latn | 13975296 | medlow | Turkic |
| 154 | Northern Kurdish | kmr | latn | 13480832 | medlow | Indo-European |
| 155 | Kabyle | kab | latn | 13282688 | medlow | Afro-Asiatic |
| 156 | Fon | fon | latn | 13019904 | medlow | Niger-Congo |
| 157 | Low German | nds | latn | 12879488 | medlow | Indo-European |
| 158 | Inuktitut | iku | cans | 12683776 | medlow | Eskimo-Aleut |
| 159 | Maithili | mai | deva | 12227712 | medlow | Indo-European |
| 160 | Lingala | lin | latn | 12203136 | medlow | Niger-Congo |
| 161 | Guarani | grn | latn | 12139904 | medlow | Tupian |
| 162 | Tibetan | bod | tibt | 12052224 | medlow | Sino-Tibetan |
| 163 | Pangasinan | pag | latn | 11895296 | medlow | Austronesian |
| 164 | Bemba | bem | latn | 11693952 | medlow | Niger-Congo |
| 165 | Wolof | wol | latn | 11647872 | medlow | Niger-Congo |
| 166 | Tumbuka | tum | latn | 11176320 | medlow | Atlantic-Congo |
| 167 | Luo | luo | latn | 11028992 | medlow | Eastern Sudanic |
| 168 | Malagasy | mlg | latn | 10417152 | low | Austronesian |
| 169 | Oromo | orm | latn | 10022016 | low | Afro-Asiatic |
| 170 | Dimli | diq | latn | 9850112 | low | Indo-European |
| 171 | Yiddish | yid | hebr | 9727872 | low | Indo-European |
| 172 | Tuvinian | tyv | cyrl | 9700736 | low | Turkic |
| 173 | Min Nan Chinese | nan | latn | 9654656 | low | Sino-Tibetan |
| 174 | Balinese | ban | latn | 9067776 | low | Austronesian |
| 175 | Fijian | fij | latn | 8515328 | low | Austronesian |
| 176 | Central Aymara | ayr | latn | 8513792 | low | Aymaran |
| 177 | Aragonese | arg | latn | 8144384 | low | Indo-European |
| 178 | Ligurian | lij | latn | 7909120 | low | Indo-European |
| 179 | Dhivehi | div | thaa | 7748608 | low | Indo-European |
| 180 | Luba-Lulua | lua | latn | 7352192 | low | Niger-Congo |
| 181 | Silesian | szl | latn | 7311872 | low | Indo-European |
| 182 | Nigerian Fulfulde | fuv | latn | 6747136 | low | Niger-Congo |
| 183 | Swiss German | gsw | latn | 6581888 | low | Indo-European |
| 184 | Swati | ssw | latn | 6076160 | low | Niger-Congo |
| 185 | Betawi | bew | cyrl | 5948160 | low | Creole |
| 186 | Friulian | fur | latn | 5731584 | low | Indo-European |
| 187 | Sardinian | srd | latn | 5723904 | low | Indo-European |
| 188 | Bavarian | bar | latn | 5696512 | low | Indo-European |
| 189 | Tok Pisin | tpi | latn | 5505792 | low | Creole |
| 190 | Umbundu | umb | latn | 5479936 | low | Niger-Congo |
| 191 | Nigerian Pidgin | pcm | latn | 5292160 | low | Creole |
| 192 | Eastern Mari | mhr | cyrl | 5290752 | low | Uralic |
| 193 | Ido | ido | latn | 4775808 | low | Constructed |
| 194 | Russia Buriat | bxr | cyrl | 4556800 | low | Mongolic |
| 195 | Bhojpuri | bho | deva | 4365440 | low | Indo-European |
| 196 | Bambara | bam | latn | 4271232 | low | Mande |

| 197 | Chokwe | cjk | latn | 4177792 | low | Atlantic-Congo |
| 198 | Southwestern Dinka | dik | latn | 4137728 | low | Nilotic |
| 199 | Dyula | dyu | latn | 3980416 | low | Mande |
| 200 | Mossi | mos | latn | 3948544 | low | Niger-Congo |
| 201 | Turkmen | tuk | latn | 3940864 | low | Turkic |
| 202 | Piemontese | pms | latn | 3818368 | low | Indo-European |
| 203 | Central Kanuri | knc | latn | 3756288 | low | Nilo-Saharan |
| 204 | Wu Chinese | wuu | hans | 3689728 | low | Sino-Tibetan |
| 205 | Kongo | kon | latn | 3668224 | low | Atlantic-Congo |
| 206 | Dargwa | dar | cyrl | 3564800 | low | Nakh-Daghestanian |
| 207 | Buginese | bug | latn | 3539840 | low | Austronesian |
| 208 | Kabuverdianu | kea | latn | 3463936 | low | Indo-European |
| 209 | Kabiyè | kbp | latn | 3286272 | low | Niger-Congo |
| 210 | Kimbundu | kmb | latn | 3169536 | low | Atlantic-Congo |
| 211 | Hawaiian | haw | latn | 2996352 | low | Austronesian |
| 212 | Sango | sag | latn | 2924928 | low | Niger-Congo |
| 213 | Mirandese | mwl | latn | 2819584 | low | Indo-European |
| 214 | Kachin | kac | latn | 2732160 | low | Sino-Tibetan |
| 215 | Ingush | inh | cyrl | 2641408 | low | Nakh-Daghestanian |
| 216 | Kikuyu | kik | latn | 2636544 | low | Niger-Congo |
| 217 | Romansh | roh | latn | 2578304 | low | Indo-European |
| 218 | Kaqchikel | cak | latn | 2560256 | low | Mayan |
| 219 | Kabardian | kbd | cyrl | 2523264 | low | Northwest Caucasian |
| 220 | Volapük | vol | latn | 2522880 | low | Constructed |
| 221 | Mandarin Chinese | cmn | hans | 2511744 | low | Sino-Tibetan |
| 222 | Kituba | mkw | cyrl | 2431872 | low | Creole |
| 223 | Magahi | mag | deva | 2379776 | low | Indo-European |
| 224 | Central Bikol | bcl | latn | 2348672 | low | Austronesian |
| 225 | Kashmiri | kas | deva | 2302592 | low | Indo-European |
| 226 | Cusco Quechua | quz | latn | 2273280 | low | Quechuan |
| 227 | Literary Chinese | lzh | hant | 2267648 | low | Sino-Tibetan |
| 228 | Walloon | wln | latn | 2234880 | low | Indo-European |
| 229 | Akan | aka | latn | 2143360 | low | Niger-Congo |
| 230 | Berber | ber | latn | 2132352 | low | Afro-Asiatic |
| 231 | Chhattisgarhi | hne | deva | 2104576 | low | Indo-European |
| 232 | Interlingua | ina | latn | 2066816 | low | Constructed |
| 233 | Upper Sorbian | hsb | latn | 2062720 | low | Indo-European |
| 234 | Latgalian | ltg | latn | 2061952 | low | Indo-European |
| 235 | Santali | sat | olck | 1973888 | low | Austro-Asiatic |
| 236 | Susu | sus | arab | 1948160 | low | Mande |
| 237 | Nuer | nus | latn | 1941760 | low | Eastern Sudanic |
| 238 | Vlaams | vls | latn | 1928064 | low | Indo-European |
| 239 | Quechua | que | latn | 1901184 | low | Quechuan |
| 240 | Udmurt | udm | cyrl | 1857664 | low | Uralic |
| 241 | Veps | vep | latn | 1844736 | low | Uralic |
| 242 | Avaric | ava | cyrl | 1772288 | low | Nakh-Daghestanian |
| 243 | Swahili | swh | latn | 1768960 | low | Niger-Congo |
| 244 | Lak | lbe | cyrl | 1715328 | low | Nakh-Daghestanian |
| 245 | Erzya | myv | cyrl | 1714432 | low | Uralic |
| 246 | Urdu | urd | deva | 1697408 | low | Indo-European |
| 247 | Ossetian | oss | cyrl | 1697024 | low | Indo-European |
| 248 | Uighur | uig | latn | 1627648 | low | Turkic |
| 249 | Lezghian | lez | cyrl | 1625344 | low | Nakh-Daghestanian |
| 250 | Goan Konkani | gom | deva | 1604096 | low | Indo-European |
| 251 | Shan | shn | mymr | 1589248 | low | Tai-Kadai |
| 252 | Serbian | srp | latn | 1543424 | low | Indo-European |

Table 2: Languages included in our language modeling study.

