# OpenReview forum: "When Is Multilinguality a Curse? Language Modeling for 252 High- and Low-Resource Languages"
_ICLR.cc/2024/Conference — Submitted to ICLR 2024_

### Official Review · Reviewer_Dd5N · 2023-10-22

**Soundness:** 3 good
**Presentation:** 3 good
**Contribution:** 3 good
**Rating:** 6
**Confidence:** 4

**Summary:**

This paper presents an extensive of controlled experiments to figure out how language performance in each language is related to (1) the size of monolingual data, (2) the added multilingual dataset size, (3) linguistic similarity of the added languages, and (4) model size. The authors pretrained more than 10,000 models over 250 languages under different settings. The authors drew various conclusions according to their experiments and showed that massively multilingual pretraining may only be beneficial for some languages, while more targeted models, e.g., monolingual models, may perform much better.

**Strengths:**

- The paper is highly motivated and is generally easy to follow.
- The authors conducted extensive and controlled experiments.
- The results are interesting and may shed light on the massively multilingual pretraining.

**Weaknesses:**

- The major weakness of the paper is that the size of all models is very small, and therefore some of the findings might not be transferred to larger models. In the experiments, the authors also show that when the model is larger, the degradation is mitigated compared with smaller models. However, it is UNREASONABLE to train many models that have the same size as, e.g., XLM-R. Therefore this is only a concern but not a suggestion to do more experiments.

- I found some parts of the papers are a bit repetitive. For example, the first and the second paragraphs in the introduction section have extensive similar claims. The authors should improve the flow of writing to reduce such redundancy.

**Questions:**

- In the paragraph "Multilingual tokenizers", the author mentioned that the target language and the added multilingual datasets are tokenized separately. And the size for multilingual vocab is set to 32K tokens, which is the same as the monolingual tokenizer. Is this a limitation since 32K tokens might not be enough to support ten languages (especially when the scripts of the ten languages are different)?

- In the very interesting part of "Syntactic similarity of added languages drives results", the authors mentioned that abstract linguistic similarity might be more beneficial than surface-level similarity. I was wondering how conceptual similarity [1] would hold in this case. Could the authors give any thought to this?

- The authors mentioned that the degradation is even slightly larger when the added languages are "similar" to the target languages. This sounds a bit counter-intuitive. Could the authors give some possible explanations?

[1] https://arxiv.org/pdf/2305.08475.pdf

---

> ### Author Response · Authors · 2023-11-15
>
> Thank you for the comments and questions!
>
> *"The major weakness of the paper is that the size of all models is very small."* See response to Reviewer HmV5.
>
> *"For example, the first and the second paragraphs in the introduction section have extensive similar claims."* Thanks for pointing this out! We have removed redundancy in those paragraphs in this revision.
>
> *"Is this a limitation since 32K tokens might not be enough to support ten languages?"* We agree that 32K tokens may not be enough to cover ten languages with high word coverage, so performance on the added ten languages may be non-optimal. For this reason, we only evaluate performance in the target language, which retains its 32K-vocab monolingual tokenizer. Increasing the multilingual vocabulary size would significantly increase computation costs due to added embedding parameters for each model.
>
> *"I was wondering how conceptual similarity would hold in this case."* The same analyses (correlations and variance partitioning; Section 6.1) could evaluate the effects of conceptual similarity of added languages on target language performance. To facilitate such evaluations, we plan to release the evaluation log-likelihood scores of all our models. Conceptual similarity is particularly interesting as an abstract similarity measure that is largely separate from structural (syntactic) similarity.
>
> *"Degradation is even slightly larger when the added languages are 'similar' to the target languages. This sounds a bit counter-intuitive. Could the authors give some possible explanations?"* We agree that this result is surprising. One possible explanation is that more similar languages affect the target language more due to surface level and representational similarities between languages. In low-resource scenarios, these influences from other languages appear to improve predictions; however, in high-resource scenarios, when models are learning more fine-grained language-specific nuances, these influences might hurt performance, making similar languages hurt performance more.

---

> > ### Comment · Reviewer_Dd5N · 2023-11-20
> >
> > Dear authors,
> >
> > Thank you for your response. My questions have been properly answered.
> >
> > I understand there is a computational budget limitation when running experiments with larger model sizes. As I said, it might be impractical and also unnecessary. Although I am still a bit skeptical if similar conclusions could be drawn for larger models (even for the XLM-R level model), I think this paper does shed light on multilingual pretraining for small-size models.

---

### Official Review · Reviewer_wTm3 · 2023-10-30

**Soundness:** 2 fair
**Presentation:** 3 good
**Contribution:** 2 fair
**Rating:** 6
**Confidence:** 4

**Summary:**

This paper evaluates the impact of multilingual pre-training on language modeling performance across 250 languages, both high-resource and low-resource. The authors find that while adding multilingual data to the training dataset can improve performance for low-resource languages, it can also hurt performance for high-resource languages. They also find that the benefits of multilingual pre-training depend on the syntactic similarity of the added languages, with marginal additional effects of vocabulary overlap. Based on the obtained results, the authors suggest that massively multilingual pre-training may not be beneficial for any languages involved, but that more targeted models can significantly improve performance.

**Strengths:**

1. Pre-train a massive number (over 10000) of monolingual and multilingual language models for over 250 low-resource and high-resource languages to provide concrete evidence for the effects of multilinguality on language modeling performance in individual languages.

2. Through the carried out experiments, this work clarifies the already-known effects of multilingual pre-training on low-resource and high-resource languages. For example, it shows that adding multilingual data improves low-resource language modeling performance, similar to increasing low-resource dataset sizes by up to 33% while the high-resource performance degradations when adding multilingual data can be similar to reducing high-resource dataset sizes by over 85%. And that the improvements depend on syntactic and lexical similarities of the added multilingual data.

**Weaknesses:**

1. I found it a strong (unsupported/contradicting) claim when the authors argue that "the multilingual pre-training may not be beneficial to any languages involved". Because the obtained results already show that low-resource languages clearly benefit from multilinguality and that the high-resource performance degradation reduces as the model size increases. Providing supportive arguments regarding this claim might be useful.

2. While the authors carried out a dramatic number of experiments, they do not provide detailed reasons why such a huge amount of computing resources would be needed. I assume selecting a set of languages that represent specific scenarios would be enough to get supportive results. And that could help in securing some resources that could be used to evaluate relatively larger language models which could have probably given a different picture of the results.

3. The performance of most LLMs is often assessed through downstream task performance since the pre-training loss cannot always fully explain downstream performance (https://proceedings.mlr.press/v202/liu23ao.html). I am wondering why the evaluation of LLMs on downstream tasks was not included.

4. In the first line of page 6, it was mentioned that a total of 8454 multilingual language models were pre-trained while in the abstract it was mentioned that over 10000 models were trained. Is that a typo?

**Minor:**
Possible formatting issue: in the current ICLR paper format, the titles of the table should be at the top of it instead of the bottom.

**Questions:**

Please, check the weakness section.

---

> ### Author Response · Authors · 2023-11-15
>
> Thank you for the helpful comments!
>
> *"Strong (unsupported/contradicting) claim when the authors argue that 'the multilingual pre-training may not be beneficial to any languages involved.'"* Thank you for pointing out this ambiguity in the original text. By this, we intend to mean that multilingual pre-training is *non-optimal* for any languages involved. Multilingual pre-training degrades high-resource language performance relative to a monolingual model (Figure 5). An excess of multilingual data (e.g. massively multilingual pre-training with many dissimilar languages) degrades low-resource language performance relative to a targeted multilingual model with a moderate amount of data from similar languages (Figure 3). Even if model sizes are dramatically increased, models optimized for many languages simultaneously would likely require more parameters than would be computationally feasible (Section 7). We will clarify these claims in the final version of the manuscript (and the abstract wording has been updated in the PDF).
>
> *"I assume selecting a set of languages that represent specific scenarios would be enough to get supportive results."* Due to the diversity of human language (e.g. Joshi et al., 2020, "The state and fate of linguistic diversity and inclusion in the NLP world"), results from a small set of languages are not always reflective of results overall. One of the main aims of this study is to create a sample of languages that better represent the diversity of the world’s languages, because we don't know how insights from diverse languages might change our inferences. Our study includes languages from 5 continents, from 29 language families, and with 30 writing systems, along with a wide variety of linguistic typological features (see WALS database; https://wals.info/).
>
> *"And that could help in securing some resources that could be used to evaluate relatively larger language models which could have probably given a different picture of the results."* Unfortunately, contemporary LLMs require extremely large amounts of compute. Because pre-training costs are linear with respect to parameter count (Deshpande et al., 2023, "Honey, I shrunk the language: Language model behavior at reduced scale"), a 1B-parameter model would take ~22x the compute of one of our largest models (45M parameters). Thus, to run 1B-parameter models, we would need to reduce our experimental conditions by about 22x (e.g. reduce languages from 252 to 11). Of course, larger models could be tested with larger compute budgets, but we note that directions of effect are consistent across the three model sizes we evaluated; for additional comments on model size, see response to Reviewer HmV5.
>
> *"I am wondering why the evaluation of LLMs on downstream tasks was not included."* We agree that this would be desirable. However, few tasks cover such a wide variety of languages and are feasible even in low-resource scenarios. For example, the massively multilingual Belebele (2023) reading comprehension dataset covers only 122 language variants. The XTREME (2020) benchmark covers only 40 languages, all of which are at least medium-low resource (i.e. not low-resource) in our study. In contrast, perplexities (and eval log-likelihoods) require no annotated data in the target language, and they are predictive of language model behavior on a variety of tasks (Xia et al., 2023, "Training trajectories of language models across scales").
>
> *"It was mentioned that a total of 8454 multilingual language models were pre-trained while in the abstract it was mentioned that over 10000 models were trained."* We pre-train 8454 multilingual models and 1989 monolingual baseline models.

---

> > ### Comment · Reviewer_wTm3 · 2023-11-20
> >
> > Thanks for addressing my concerns.
> >
> > I have updated the scores accordingly.

---

### Official Review · Reviewer_yYRw · 2023-11-06

**Soundness:** 3 good
**Presentation:** 3 good
**Contribution:** 2 fair
**Rating:** 5
**Confidence:** 3

**Summary:**

This paper trains up a huge number of monolingual and multilingual language models to study the effect of multilingual pretraining on language modelling performance. They covered over 250 languages and systematically vary monolingual dataset size, multilingual dataset size, linguistic similarity and model size. Their conclusions were already common knowledge. They show that adding multilingual training data improves low-resource language performance. They show that multilingual training hurts high-resource languages. They show that language similarity matters, more similar being better of course. Possibly more novel is that they show that for language similarity syntactic similarity matters - which is unexpected. But by syntactic similarity they mean the syntactic component of lang2vec which is taken from the WALS typological database so these are syntactic features collected by linguists not syntactic trees over the language.

Overall the paper was very ambitious with the scale of the experiments (trained over 10000 language models - even if they were all very small up to 45M paramteres).  The problem was that I didn't learn much.

**Strengths:**

Thorough explorations of the experimental conditions
They come up with a way of measuring model performance which is comparable across the different languages: estimated model tokens measure.

**Weaknesses:**

Unsurprising results
The models themselves were all very small - up to 45M parameters. It is not clear if these conclusions still hold for larger models as LLMs tend to be in billions of parameters and scale has been shown to be crucial for various important abilities like zero-shot and in-context learning.

**Questions:**

Why do you think that selecting most related vs least related had relatively little impact?

---

> ### Author Response · Authors · 2023-11-15
>
> Thank you for the feedback!
>
> *"Their conclusions were already common knowledge."* We agree with the reviewer that there is previous evidence for the curse of multilinguality in language models (and evidence of benefits for low-resource languages; Section 2), but it is limited in generality. Previous findings are based on relatively few scenarios. For example, XLM-R considers only five subsets of pre-training languages for downstream tasks in 16 languages. Other studies (e.g. mBERT, BLOOM, and XGLM) simply pre-select a set of languages and fix a sampling parameter to upsample low-resource languages. Yet, despite the lack of controlled studies evaluating the effects of multilinguality on language modeling performance, language models are increasingly pre-trained on multilingual data (e.g. GPT-4 and PaLM 2).
>
> Thus, our work makes several novel contributions:
> 1. It contributes a controlled method, a new measure (estimated monolingual tokens), and experimental manipulations that quantify effects of multilingual pre-training in ways that are comparable across languages and informative for pre-training in practice.
> 2. It quantifies these effects in over 250 languages. We plan to release the evaluation log-likelihood scores of all our models.
> 3. It confirms and quantifies several effects that are commonly assumed (i.e. we estimate comparable dataset sizes for low-resource performance improvements and high-resource performance degradations, for similar and dissimilar languages).
> 4. It uncovers several results that may be unexpected: syntactic similarity plays a greater role than vocabulary overlap in low-resource performance improvements, low-resource performance improvements decrease given excessive multilingual data, and more similar languages degrade high-resource performance more than dissimilar languages (see next point).
>
> *"They show that language similarity matters, more similar being better of course."* Surprisingly, adding data from similar languages hurts performance more than adding data from dissimilar languages for high-resource language performance (Section 6.2). This may be because more similar languages affect the target language more due to surface level and representational similarities. In low-resource scenarios, these influences from other languages improve predictions; however, in high-resource scenarios, when models are learning more fine-grained language-specific nuances, these influences might hurt performance, making similar languages hurt performance more.
>
> *"These are syntactic features collected by linguists, not syntactic trees over the language."* Syntactic trees are not available for the majority of languages in our study. For example, the Universal Dependencies dataset contains only 74 language varieties with at least 1K sentences. Therefore, we turn to typological databases, which have data for the vast majority of the languages in our study. An additional benefit of using a typological database like WALS is that it allows us to label each language according to specific syntactic features in a controlled and comparable way. In linguistics, this is the gold standard for comparing typological features in a large number of languages.
>
> *"The models themselves were all very small - up to 45M parameters."* See response to Reviewer HmV5.
>
> *"Why do you think that selecting most related vs least related had relatively little impact?"* We agree that the effects of language similarity are small in many, although not all (e.g. Figure 3, top right), cases. Some possible explanations are:
> * Effects of multilinguality may be driven by linguistic similarities not captured well by our similarity metric (i.e. not syntactic, geographic, or lexical similarities). For example, see conceptual similarity (suggested by Reviewer Dd5N).
> * The benefits of multilinguality may be based on fairly abstract linguistic properties that are shared to a large degree across most or many languages, regardless of relatedness (e.g. hierarchical structure in general). The drawbacks of multilinguality may be based on nuances that are highly language-specific, regardless of linguistic relatedness or similarity.
> * We do not have closely related languages for some languages (particularly some low-resource languages), which might make the similar languages condition closer to the dissimilar languages condition for those languages.

---

### Official Review · Reviewer_HmV5 · 2023-11-07

**Soundness:** 4 excellent
**Presentation:** 4 excellent
**Contribution:** 3 good
**Rating:** 8
**Confidence:** 4

**Summary:**

This paper investigates the effects of multilinguality in language model training. The authors take 250 languages, each with varying amounts of textual data, and train GPT2-style models in three sizes. For each, they study the effects of injecting multilingual data (of either related or unrelated languages) in various amounts. The findings are that, for low-resource languages, adding moderate amounts of multilingual data can aid target language performance, as long as model capacity is sufficient. Whereas for high-resource languages, multilingual training is consistently hurtful in the settings evaluated by the authors.

**Strengths:**

* An in-depth study of the effects of multilingual pretraining for small-ish language models. Ablation studies are solid, covering resource level, varying the amount of multilingual data that's injected, similar/dissimilar languages, the effects of syntactic/lexical/geographic similarity, and model size.
* Overall, this paper sheds light on the degree to which cross-lingual transfer can help. There is already evidence coming from the field of machine translation showing that high-resource language performance suffers in multilingual settings (e.g. by comparing the performance of languages such as English or Mandarin in massively multilingual MT models such as NLLB, arXiv:2207.04672, vs equivalent bilingual models). The authors of this paper take this a step further by providing a comprehensive evaluation across many languages and model sizes.
* The carefully chosen tokenization approach (section 5) and evaluation metric (section 4.3) allow for intuitive comparison of monolingual and multilingual models.

**Weaknesses:**

* The model sizes are all on the small end. One can see from e.g. Figs. 3 and 5 that, as model size increases, multilingual pretraining hurts performance less and less (and actually becomes beneficial, in a few cases). The obvious next step would then be to grow model size further. I fully acknowledge, however, that this kind of experiment would have prohibitive computational costs, and I believe that the paper is already valuable as-is. I further appreciate the authors explicitly calling out and addressing this limitation in sec. 7.1.

**Questions:**

Do you expect these results to hold also for related tasks such as machine translation, speech recognition, etc.? Based on the current SOTA models for these related tasks, would you agree that the story is likely to be similar there too?

---

> ### Author Response · Authors · 2023-11-15
>
> Thank you for the comments!
>
> *"The model sizes are all on the small end."* Our experiments take approximately 17700 A6000 GPU hours total, which is still less than 1/1500x the FLOPs required to train the original 175B-parameter GPT-3 model (computational costs added to Appendix A.3). Other research groups may have the computational resources to extend our experiments to larger models, but we believe that the sizes of our models are a reasonable compromise between informativity of results and environmental/computational costs. Our results still cover three model sizes, 250 languages, and four experimental manipulations. Directions of effect are consistent across the three model sizes we evaluated.
>
> The added benefits of running similar experiments for larger models may also be limited. As dataset sizes increase to hundreds of billions of tokens, it is likely that larger models will still hit multilingual capacity limitations (e.g. 70B-parameter models are required just for English; Hoffmann et al., 2022, "Training compute-optimal large language models"). Furthermore, smaller models can perform comparably to larger models in low-resource scenarios (Section 7.1), making them useful for efficient low-resource language technologies and low-compute settings such as laptops and mobile phones. Nevertheless, results for larger models could be verified with larger compute budgets.
>
> *"Do you expect these results to hold also for related tasks such as machine translation, speech recognition, etc.?"* Although we cannot guarantee the same results for other tasks, we believe that results may be similar for tasks with similar setups. For example, tasks such as translation still use Transformer text generation models, although often with encoder-decoder architectures. Like in language modeling, massive multilinguality in machine translation appears to benefit low-resource languages (NLLB team, 2022, "No language left behind: Scaling human-centered machine translation"; Bapna et al., 2022, "Building machine translation systems for the next thousand languages"), but the benefits and drawbacks of massive multilinguality are rarely quantified using controlled studies. Results for less similar tasks and architectures (e.g. speech models) would need to be verified empirically. In any case, multilingual language modeling over text (e.g. for text generation; GPT-4, PaLM 2, BLOOM, XGLM) is increasingly popular as a task, with direct implications for downstream applications such as chatbots and virtual assistants.

---

### Meta-Review · Area_Chair_tMyh · 2023-12-10

**Metareview:**

This paper presents a study of the effect of combined multilingual pretraining of the models on the monolingual performance for low, medium and high resource languages. The authors conclude that adding multilignual data helps low resource language modeling, but adversely affects high resource language modeling. Further as more multlingual data is added both low and high resources languages' modeling is hurt. I find the former part of the results quite obvious, but I'm also surprised about latter part. However this could be expected because the model size being used is quite small (45M), and hence the learning capacity of the model is low.

The paper mainly suffers from two issues: (1) there is no downstream application evaluation on the abilities of the pre-trained models. Even if such eval benchmarks are available only for 40 languages as authors noted, this evaluation would be desirable. (2) the size of the model is so small, that its unclear what pre-training here really means. The finetuning of these models for downstream tasks could totally wipe out the effect of pre-training and hence any of the conclusions drawn here could be meaningless. In general finetuning is needed when the parameter space is way too large to be trained with just finetuning datasets.

I would recommend that the authors reduce the number of languages if required to get more computation and perform the above experiments on a smaller set of languages so that usable conclusions can be drawn.

**Justification For Why Not Higher Score:**

More empirical analysis is needed to justify the utility of the conclusions.

**Justification For Why Not Lower Score:**

n/a

---

### Decision · Program_Chairs · 2024-01-16

Reject